# Natural Killer Cells: A Promising Kit in the Adoptive Cell Therapy Toolbox

**DOI:** 10.3390/cancers14225657

**Published:** 2022-11-17

**Authors:** Jiani Xiao, Tianxiang Zhang, Fei Gao, Zhengwei Zhou, Guang Shu, Yizhou Zou, Gang Yin

**Affiliations:** 1Department of Pathology, School of Basic Medical Sciences, Xiangya Hospital, Central South University, Changsha 410000, China; 2Department of Immunobiology, Yale University School of Medicine, New Haven, CT 06511, USA; 3Department of Immunology, School of Basic Medicine, Central South University, Changsha 410000, China; 4National Clinical Research Center for Geriatric Disorders, Xiangya Hospital, Central South University, Changsha 410000, China

**Keywords:** NK cells, adoptive cell therapy (ACT), cancer immunotherapy, chimeric antigen receptor (CAR), genetic engineering

## Abstract

**Simple Summary:**

Adoptive cellular immunotherapy is one of the most effective strategies promising to eliminate tumors. Natural killer (NK) cells gain increasing attention for their stable availability and safety properties. To improve the anti-tumor efficacy of NK cells, multiple genetically engineered NK cellular products have been tested in recent years. In this review, we evaluate the latest research progress on adoptive NK cell therapy with a special focus on their alternative sources and genetic modifications.

**Abstract:**

As an important component of the innate immune system, natural killer (NK) cells have gained increasing attention in adoptive cell therapy for their safety and efficacious tumor-killing effect. Unlike T cells which rely on the interaction between TCRs and specific peptide-MHC complexes, NK cells are more prone to be served as “off-the-shelf” cell therapy products due to their rapid recognition and killing of tumor cells without MHC restriction. In recent years, constantly emerging sources of therapeutic NK cells have provided flexible options for cancer immunotherapy. Advanced genetic engineering techniques, especially chimeric antigen receptor (CAR) modification, have yielded exciting effectiveness in enhancing NK cell specificity and cytotoxicity, improving in vivo persistence, and overcoming immunosuppressive factors derived from tumors. In this review, we highlight current advances in NK-based adoptive cell therapy, including alternative sources of NK cells for adoptive infusion, various CAR modifications that confer different targeting specificity to NK cells, multiple genetic engineering strategies to enhance NK cell function, as well as the latest clinical research on adoptive NK cell therapy.

## 1. Introduction

Recently, immunotherapy has attracted broad attention worldwide as an innovative approach of cancer treatment. The core of the immunotherapy strategy is to eliminate tumor cells by enhancing the patient’s immune system. As an important part of the immune system, the number and activity of immune effector cells, especially natural killer (NK) and T-cells, are closely related to the body’s anti-tumor ability. Hence, T or NK-based adoptive transfer strategy has provided new hope for patients with advanced malignant tumors including lymphoma [1,2], leukemia [3], melanoma [2], lung cancer [4], and colorectal cancer [5].

In the 1980s, Rosenberg and his group were the first to confirm that treating metastatic melanoma patients with autologous tumor-infiltrating lymphocytes (TILs) and IL-2 could achieve objective regression [6]. Since then, immunotherapeutic approaches based on adoptive cell transfer have advanced significantly with increasing sources of effector cells [7], as well as tools to enhance specificity, such as engineered specific T-cell receptors (TCR) [8], and chimeric antigen receptors (CAR) [9]. As the “first line of defense” of the human immune system, NK cells can directly recognize and rapidly kill target cells without antigen pre-sensitization and major histocompatibility complex (MHC) restriction. Compared with T cells, one of the important characteristics of NK cells is the lack of TCRs (one of the main mechanisms of allogenic T-cell activation) [10,11]. Inhibitory receptors of NK cells, such as KIR, also prevent NK cells from attacking normal host cells. Hence, NK cells are considered safer for the ACT due to much fewer possibilities of inducing graft versus host disease (GvHD). Moreover, these features of NK cells make it possible that they can be provided by any healthy donor, which breaks the limitation of T-cell-based ACT and reduces costs for large-scale production [12]. These features make NK cells an ideal candidate to become an “off-the-shelf” therapy product for adoptive immunotherapy. Specifically, autologous or allogeneic NK cells are expanded and activated under the stimulus of feeder cells, such as gene-modified K562 cells and cytokines (IL-2/IL-15) in vitro, and then infused back into patients to eliminate tumor cells in vivo through direct cytotoxicity and cytokine secretion [13,14]. Such conventional NK-based immunotherapy has shown preliminary benefits in various malignancies, especially hematological malignancies, together with limited toxicity against normal tissues [15,16,17]. Despite this, several obstacles still restrict the widespread application of NK cells in vivo, including low cytotoxicity, insufficient persistence, and inertness in the tumor microenvironment (TME) [18,19]. Encouragingly, the rapidly evolving field of adoptive NK cell therapy now encompasses a series of candidate cell sources [20], and advances in genetic engineering have further improved the specificity, persistence, and anti-tumor efficacy of NK cells [21].

In this review, we first described the basic characteristics and major functions of NK cells. We then reviewed possible sources of NK cells and ways of genetic modifications on NK cells for adoptive immunotherapy. We focused on the current application of gene-editing technologies in adoptive NK cell therapy, especially the progress made in the field of CAR NK-based therapy. Furthermore, we summarized the latest clinical results in the adoptive NK cell therapy for patients with malignancies. Finally, we propose the future directions of NK-based adoptive cell therapy (ACT).

## 2. An Overview of NK Cells

### 2.1. Biological Properties of NK Cells

NK cells are important immune effector cells and belong to the innate lymphoid cells (ILCs) family. NK cells were originally named for their natural killer activity against cancer cells [22]. During the development process, NK cells are derived from the same common lymphoid progenitor cells (CLPs) as T-cells and B-cells. CD34^+^ hematopoietic progenitor cells (HPCs) migrate from the bone marrow (BM) to numerous anatomical sites, including the thymus, lymph nodes, spleen, liver, etc. Through different developmental stages, HPCs drive the differentiation and maturation of NK cells by gradually down-regulating CD34 and up-regulating CD56 [23]. According to the difference in the expression density of CD56 and CD16, human NK cells can be further divided into two well-characterized subsets: CD56^bright^ CD16^−^ and CD56^dim^CD16^+^. The former predominantly reside in secondary lymphoid tissues, and their function is mainly to produce cytokines, including interferon-γ (IFN-γ), tumor necrosis factor (TNF), IL-10, and granulocyte-macrophage colony-stimulating factor (GM-CSF) [24,25]. In contrast, the CD56^dim^CD16^+^ subset accounts for about 90% of all NK cells in peripheral blood (PB) [26], mediating higher baseline perforin expression and stronger natural cytotoxicity [20].

Although NK cells lack somatic rearrangement receptors that specifically distinguish antigens, they inherently express a diverse repertoire of activating, co-stimulatory and inhibitory receptors. The dynamic balance between these receptors ensures a fine regulation of NK cell cytotoxic activity (Figure 1). NK cells are activated when the activation signal is stronger than the inhibitory signal; otherwise, they are in a tolerant state. Aggregation of activating receptors induces the activation of associated Src family kinases, which further phosphorylate the conserved tyrosine in the immunoreceptor tyrosine-based activation motifs (ITAM) and initiate activating signaling cascades [27]. Activating receptors include Fcγ receptor CD16, C-type lectin family (NKG2D, NKG2C), natural cytotoxicity receptors (NKp30, NKp44, and NKp46), cytokine-binding receptors, and many others [28]. These receptors, after binding with cognate ligands, induce polyfunctional NK cells degranulation and cytokine secretion (IFN-γ and MIP-1β) [29]. Killer cell immunoglobulin-like receptors (KIRs) and NKG2A/CD94 heterodimers are widely studied NK cell inhibitory receptors that recognize HLA class I molecules [20]. Immunoreceptor tyrosine-based inhibition motifs (ITIMs) in the intracellular domains act as an interface between these receptors and inhibitory signaling pathways, mediating the attenuation of NK cell activation signals [30].

### 2.2. Effector Function of NK Cells

#### 2.2.1. Anti-Tumor Function

NK cells can initiate the cancer-immune defense function for the first time and respond rapidly to abnormal (pathological or malignant) cells. As a “warrior” with potent anti-cancer activity, NK cells function mainly by direct killing, antibody-dependent cell-mediated cytotoxicity (ADCC), and secreting cytokines.

In the early stages, tumor cells selectively lose MHC class I molecules to evade the T-cell attack [31]. However, these cells become sensitive to NK cell-mediated elimination because they have lost the NK cell inhibitory signal simultaneously. Additionally, cells with up-regulated expression of activating ligands are also targets of NK cells. NK cells recognize the corresponding ligands on the surface of these sensitive cells and directly kill tumor cells through non-specifically releasing killing mediators (perforin and granzyme) or by binding to death receptors (Fas-FasL and TRAIL-TRAILR) [32]. NK cells can also secrete extracellular vesicles (EVs), which contain nucleic acids, EV proteins, and cytolytic molecules that ultimately result in the apoptosis of tumor cells without cell-to-cell contact [33,34].

Another powerful anti-tumor function of NK cells is ADCC, predominantly mediated by CD16 binding to the Fc segment of antibodies [35]. Various clinically developed monoclonal antibodies (mAbs), such as rituximab [36], cetuximab [37], and infliximab [38], have been suggested to exploit this mechanism of NK cells to exert their anti-tumor effects. Moreover, activated NK cells also interact with other immune cells by synthesizing and secreting multiple cytokines, including IFN-γ, TNF, lymphotactin (XCL1), CC-chemokine ligand (CCL3, CCL4, and CCL5), IL-10, IL-13 and GM-CSF [39,40,41,42], thereby regulating the body’s immune function [41,43]. For example, secretion of IFN-γ induces T helper 1 cell polarization and stimulates tumor-specific CD8^+^ T-cells to enhance adaptive immunity [39,44]. Secretion of CCL5, XCL1 and FLT3L recruits dendritic cells to the tumour microenvironment and invokes proliferation and activation of these cells [41,42].

#### 2.2.2. Pro-Tumor Function

However, NK cells also possess plasticity, which can be polarized to a pro-tumor type in response to the TME signals, as well as promote tumor angiogenesis and metastasis [45,46].

Increasing studies have found the infiltration of decidual NK-like cells in a variety of tumor tissues, including lung cancer [47], renal cell carcinoma [48], colorectal cancer [49], breast cancer [50], and melanoma [51]. Decidual NK cell is a unique NK cell subset to drive vascularization during embryo development [45]. Decidual NK-like cells possess the same CD56^bright^CD16^−^ phenotype and pro-angiogenic activities as decidual NK [52]. For example, a study has shown that supernatants derived from non-small cell lung cancer (NSCLC) decidual NK-like cells can induce the formation of capillary-like structures in vitro [47]. During immunoediting, malignant cells evolve various strategies to evade NK cell attack and form macrometastases [53], such as up-regulating inhibitory ligands and down-regulating activating ligands [54,55], secreting metalloproteinases that mediate shedding of activating ligands [56,57], or reprogramming NK cells to a metastasis-promoting cell state [58]. NK cells also exist immune exhausted status under tumors or chronic infections. These cells often express markers of cell exhaustion, such as PD-1 [59,60], CD96 [61], TIGIT [62], NKG2A [63], and Tim-3 [64]. Moreover, their ability to produce cytotoxic factors (IFN-γ, granzyme, perforin) decreased significantly [63,65].

Soluble factors and cell-to-cell interactions within TME are the main reasons for inducing NK cell polarization and exhaustion [45,52,65]. Many immunosuppressive soluble factors, including TGF-β [66,67], prostaglandin E2 [68], human leukocyte antigen G (HLA-G) [69], hypoxia [67], and glycodelin-A [70], have been confirmed to induce NK cells towards a pro-tumorigenic phenotype. Moreover, several TME-associated cells, such as tumor-associated macrophages (TAMs), myeloid-derived suppressor cells (MDSCs), and regulatory T-cells (Tregs) could dampen NK cell cytotoxic activities by direct contact and secretion of soluble factors TGF-β and prostaglandin E2 [45,60,71,72] (Figure 1).

Given that these interactions could compromise the activity of NK cells, immunotherapy based on NK cells needs to consider the inherent plasticity of NK cells and design optimal strategies to promote effective phenotypes or limit the polarization to pro-tumor phenotypes.

## 3. NK-Based ACT

As the most commonly used immune effector cells in cellular immunotherapy, T-cells have achieved gratifying clinical efficacy in treating B-Cell Lymphoma [73,74]. CARs are artificial cell surface activating receptors that recognize specific antigens expressed on the surface of tumor cells. The clinical success of CAR-T therapy is booming in the adoption of cellular infusion [75,76,77]. To date, four CAR-T-cell products targeting CD19 and one targeting BCMA have been approved by the FDA for application [78,79,80,81]. Despite this, the field of CAR-T therapy still faces many challenges: (1) high levels of pro-inflammatory cytokines predispose to immune cell-associated neurotoxicity (ICANS) and cytokine release syndrome (CRS) with severe life-threatening toxicity [82,83,84]; (2) HLA barriers limit the cell source for large-scale production; (3) antigen escape induces treatment resistance and relapse; (4) limited migration and invasion in solid tumors [85,86]. NK cells have gained increasing attention for their advantages such as stable availability and safety properties [87]. The main preparations are similar between NK-based and T-based adoptive immunotherapy. NK cells are collected from patients or healthy volunteers typically and undergo a series of processes in vitro, including purification, activation and expansion, quality control, and eventually infusion into patients. These NK cell products not only kill target cells directly in vivo but also control cancer recurrence and metastasis by activating and/or enhancing the body’s immunity. In recent years, researchers have focused on optimizing the source and function of NK cells, especially with the advances in genetic engineering, making NK cells a unique and promising therapeutic tool in the ACT field (Figure 2).

### 3.1. Cell Sources

To generate sufficient and high-quality NK cells, cell sources for application in immunotherapy have been largely enriched, including PB-NK [14,88,89], umbilical cord blood NK (UCB-NK) [90,91,92], NK cell lines [93,94,95,96], and stem cell-derived NK [97,98] (Figure 2).

#### 3.1.1. PB-NK

Autologous or allogeneic PB is the most commonly used cell source in NK ACT. A phase I clinical trial indicated that autologous NK cells (expanded and activated) could exhibit preliminary anti-tumor activity on HER2-positive malignancies via trastuzumab-mediated ADCC [99]. However, for cancer immunotherapy, autologous NK cells are often exhausted or impaired by self-HLA class I levels [100]. In contrast, allogeneic PB-NK showed a stronger tumor-killing ability due to KIR-HLA incompatibility [101].

Notably, a unique cell subset in the PB of healthy individuals with prior exposure to human cytomegalovirus, termed FcRγ-deficient NK (g-NK) cells, is characterized by deficient for the signaling adaptor FcRγ and normal CD3ζ expression [102]. Compared with conventional NK cells, g-NK cells display a weaker direct killing effect on target cells but show stronger secretion of cytokines (IFN-γ and TNF-α) to CD16 stimulation [103]. Interestingly, Zhang and colleagues further found that g-NK cells presented distinct adaptive immunity features and also have a dominant advantage over memory-like T-cells in number [102]. In addition, g-NK cells exhibit robust amplification on virus-infected cells in an antibody-dependent manner and maintain a constant level in vivo for a long time [104]. In multiple myeloma (MM), the latest preclinical trial has confirmed that when combined with daratumumab, expanded g-NK cells with low CD38 expression and potent ADCC activity presented dramatically superior curative and persistence than conventional NK cells [105]. These characteristics of long-lived g-NK cells suggest that they have the potential to become a novel cell source in adoptive immunotherapy to provide more powerful therapeutic effects.

Under a series of cytokine stimulation [106], NK cells demonstrate memory-like properties similar to T-cells and B-cells. PB-NK cells preactivated briefly by the combination of IL-12, IL-15, and IL-18, show a stronger IFN-γ response upon restimulation of cytokine or K562 leukemia cells [107]. As a promising cell source in ACT, cytokine-induced memory-like NK (CIML-NK) has achieved great success in the anti-tumor response of NK-resistant lymphoma and melanoma and effectively induced complete remission in some leukemia patients [108,109,110].

#### 3.1.2. UCB-NK

Unlike PB-NK, NK cells in UCB are more abundant (30% of lymphocytes), proliferative, and easy to collect. The current platform has been able to expand highly pure (>99.9%) clinical-grade UCB NK cells within 2 weeks for cancer immunotherapy [111]. After long-term cryopreservation, the expansion efficiency and cytotoxicity of UCB-NK cells are intact [92,112]. Compared with PB CD56^dim^ cells, resting UCB CD56^dim^ cells overexpressing pro-apoptotic genes are at an earlier developmental stage [113]. They typically express higher levels of NKG2A and NKG2D and functionally manifest reduced cytotoxicity in vitro against multiple non-Hodgkin’s lymphoma [113]. Of note, after stimulation with cytokines (IL-2 and IL-15), UCB-NK significantly upregulates the expression of NKp46, achieving lysis activity comparable to that of equally treated PB NK cells [114]. When cultured with γ-irradiated PLH feeder cells, UCB-NK cells were observed to have greater cytotoxicity than PB-NK cells against primary MM cells [90]. Moreover, the combination with antibody therapy also facilitates the infiltration of UCB-NK cells in the tumor bed in immunodeficient mice [91].

#### 3.1.3. NK Cell Lines

NK cell lines, such as NK-92, KHYG-1, YTS, HANK-1, and NKL, are attractive sources for NK-based immunotherapy. The major advantage of NK cell lines is that they can be easily cultured, expanded, and genetically edited. NK-92, isolated from a 50-year-old male patient with non-Hodgkin’s lymphoma, has been tested in phase I clinical trials [95,115,116,117]. Since NK-92 is tumor-derived, it must be irradiated before administration to ensure its safety. NK-92 has also been genetically engineered with CAR to treat a variety of cancers in preclinical studies, including melanoma, high-risk rhabdomyosarcoma, and lymphoma [118,119,120]. Recently, NK-92 cells have been developed to express chimerically modified CD16 receptors to enhance the lysis efficacy against tumor cells [121,122]. Compared with CAR NK derived from a primary human donor, CAR NK-92 cells secrete more granzyme A and IL-17A. Their stronger cytotoxicity against leukemia cells also leads to the destruction of healthy cells [123]. A modified version of NK-92, NK-92MI (IL-2-independent) has been explored in various cancers treatment for its excellent expansion ability and cytotoxicity [96,124,125,126]. Notably, NK101 (from an NK/T-cell lymphoma patient) has the potential to produce immunostimulatory cytokines to mobilize host anti-tumor immunity, which is the first cell line to show superior cytotoxicity to NK-92 in the 4T1 mammary carcinoma model [93]. Terminally irradiated NK cell lines often require multiple repeat infusions with non-viable cells to achieve persistence in vivo, but there also has some concerns, such as the potential for alloimmunity and rejection.

#### 3.1.4. Stem Cell-Derived NK

Another ideal candidate for “off-the-shelf” product development is stem cell-derived NK cells. Induced pluripotent stem cell (iPSC)-derived NK cells have several advantages, such as yielding a homogenous population of NK cells, an unlimited source of NK cells, and strong plasticity [127]. Although NK cells from different iPSC lines represent the differential expression of KIRs, their cytotoxicity is generally similar [128]. iPSC-NK cells are typically used for gene editing at the clonal level to improve the anti-tumor activity of NK cells [97,98].

By and large, the expansion of PB-derived NK cells to adequate cell doses is labor- and time-consuming in trials to date. NK cell lines may induce cell toxicity and rejection. In contrast, UCB or iPSC-derived NK cells are capable of adequate ex vivo expansion and appear to be safe, demonstrating the novel therapeutic potential for future clinical application.

### 3.2. Genetic Engineering

Genetic engineering technology generally refers to a series of artificial modifications of genes to create new genetic characteristics. Recently, multiple genetic engineering approaches have been tested to optimize the anti-tumor activity of NK cells and provide better adoptive NK cell therapy for cancer patients. Currently, genetic engineering on NK cells mainly includes arming NK cells with CAR or other constructions to improve the efficacy and specificity of NK cells (Figure 3 and Figure 4).

#### 3.2.1. CAR-Modified NK Cells

NK cells are a promising alternative kit in the ACT toolbox. First, NK cell products are clinically proven to be safe, and their toxicity is mild [13,129]. Second, NK cells have abundant sources with fewer graft restrictions. Third, NK cells exert CAR-dependent targeting specificity while retaining their original natural cytotoxicity, thereby helping to overcome antigen escape [130]. In addition, a variety of genetic modification strategies have further enhanced the specific and non-specific cytotoxicity, homing and infiltration capacity, in vivo persistence, and resistance to immunosuppression of CAR NK. Depending on the number of intracellular activation signals, conventional CAR receptors can be divided into three generations (Figure 3). Other more advanced CAR structures are being designed and tested to optimize the therapeutic efficacy against cancer, such as increasing the expression of cytokines [131], simultaneously targeting multiple specific antigens [132,133,134], using a construct based on nanobodies with high stability and solubility [96,135], or introducing a novel chimeric costimulatory converting receptor [136], an NK cell costimulatory domain [137,138], and a suicide gene that control the number of infused NK cells to avoid toxic side effects [139]. In the following, we presented some recent preclinical studies of CAR-NK cells (Table 1).

##### CAR NK Cells against Hematological Malignancy

As a molecule widely and specifically expressed in B-cell malignancies, CD19 serves as an ideal target for the ACT. One of the earliest studies showed that primary NK cells bearing a second-generation anti-CD19 CAR could overcome inhibitory signaling and specifically kill leukemic cells [161]. In this study, the investigators also found that co-culture with K562-mb15-41BBL favors the expansion and retroviral transduction of NK cells in vitro. CD3ζ fragment in the anti-CD19 CAR has induced superior NK-cell activity compared with DAP10, while the NK-cell activation and cytotoxicity can be further enhanced by combining both 4-1BB and CD3ζ [161]. However, after anti-CD19 CAR T/NK cell therapy, some leukemia cells develop genetic mutations and loss of CD19 antigen heterozygosity, resulting in relapsed or refractory acute leukemia [162]. This has spurred the progression of CAR NK cells that recognize alternative B-cell acute lymphoblastic leukemia (B-ALL) targets, such as FMS-like tyrosine kinase 3 (FLT3). Oelsner et al. confirmed that FLT3-positive leukemia cells specifically activated NK-92 cells expressing an anti-FLT3 CAR [139]. Interestingly, researchers also use *inducible caspase 9* (*iCasp9*), a suicide gene that induces the rapid apoptosis of transduction cells, as a safety switch rather than traditional γ-irradiation for the CAR NK-92 cells to prevent possible adverse events [139].

CD38, a promising target for cytotoxic Abs therapy, is commonly overexpressed in MM and other hematological malignancies [163]. Nanobodies are single immunoglobulin variable domains derived from camelids’ heavy-chain antibodies, which exhibit low immunogenicity, high specificity, and stability [164]. A third-generation CAR construct based on nanobodies has been developed to target three different epitopes of CD38 [135]. These CAR NK-92 cells possess potent and specific lytic activity against CD38-expressing Burkitt lymphoma cell lines and primary MM cells in vitro, providing a basis for future clinical development [163]. In addition, to prevent CD38 expression on normal NK cells from impairing the efficacy of CAR NK treatment, researchers have utilized CRISPR/Cas9 genome editing to disrupt the expression of CD38 on primary NK cells and then combined mRNA electroporation technology to bring an affinity-optimized CD38 CAR [140]. They finally construct endogenous CD38 knockdown CAR NK cells with promising success in minimizing NK cell fratricide and specifically killing CD38^+^ acute myeloid leukemia (AML) cell lines and primary leukemia cells [140].

The application of CAR T cells in T-cell acute lymphoblastic leukemia(T-ALL) therapy remains restricted due to some shared antigens between effector cells and malignancies. Such as CD5 and CD7, important characteristic markers of malignant T cells, which are also commonly expressed on normal T cells. Preclinical research has revealed that anti-CD5 CAR NK-92 cells could be an alternative therapeutic strategy, and in this study, the intracellular part of 2B4, an NK-related costimulatory receptor, was integrated into the anti-CD5 CAR construct. In contrast to the T-cell-associated costimulatory domain 4-1BB, the 2B4 costimulatory domain promoted NK cells’ proliferation and their cytotoxic activity against CD5^+^ hematological malignant cells. The ability of anti-CD7 CAR NK-92MI to specifically eradicate CD7-positive tumor cells was observed both in vitro and in T-ALL PDX mouse models [96]. In addition, bivalent anti-CD7 nanobody sequence CAR NK-92MI displayed robust IFN-γ and Granzyme B secretion against primary T-ALL samples [96].

Recently, immunotherapies targeting CD33 and CD123 have achieved promising results in treating AML. The latest study showed that CD33-targeting CAR NK cells combined the broad expression advantage of CD33 targets and the safety of NK cells, effectively eliminating the engraftment of CD33^+^ AML cells in the BM and spleen [141]. In a different study, NK-92 cells stably expressed anti-CD123-CAR and IL-15 [142]. CAR-modified NK-92 cells were significantly more cytotoxic to CD123^+^ AML cells than unmodified NK-92 cells both in vitro and in an AML-PDX model [142]. Another study tested 8 different CAR structures that contain scFv targeting CD123 and 4 different components (DAP10, FcεRIγ, 2B4, and the ζ chain of the T-cell receptor) on PB-NK cells [143]. Among various combinations, the construct CD123-2B4-CD3ζCAR NK cells not only possessed superior target cell-specific lysis ability in vitro but also had the most pronounced tumor growth inhibitory effect in the AML xenograft model [143]. Furthermore, transgenic co-expression of IL-15 endowed CAR NK cells with enhanced activation and long-term cytolytic activity. However, it also showed lethal toxicity in mouse models [143]. These results suggest that 2B4 is a more suitable component of CAR NK, although the application of IL-15 can improve the persistence of NK, it is also essential to ensure its safety in clinical application.

##### CAR NK Cells against Solid Malignancy

Solid tumor-related antigens have also been tested as targets for NK-based immunotherapies. For example, Glypican-3 (GPC3), a tumor-specific antigen, is highly expressed in hepatocellular carcinoma (HCC), hepatoblastoma, and clear-cell carcinoma of the ovary (CCCO). A GPC3-specific CAR was designed and delivered into NK-92 cells to generate a novel CAR-NK cell line termed NK-92/9.28.z [148]. In vitro, NK-92/9.28.z cells exhibited specific cytotoxicity against GPC3^+^ HCC cells with high levels of secretion of IFN-γ, and the function of NK-92/9.28.z cells was not affected by inhibitory factors, such as hypoxic, soluble TGF-β and GPC3 [148]. In vivo, irradiated NK-92/9.28.z cells were also demonstrated to be enriched at the tumor site and efficiently inhibited the growth of GPC3^+^ HCC xenografts without severe adverse effects [148]. Moreover, the introduction of the NK-cell costimulatory domain (DNAM1 and 2B4) further optimized the GPC3-specific CAR construction, allowing CAR NK-92 cells to exhibit stronger proliferation and anti-apoptotic ability and achieve predominate cytotoxicity against HCC cells in vitro [138]. To make therapeutic NK cells more homogeneous, Ueda et al. established a master cell bank by selecting HLA homozygous iPSC clones with anti-GPC3 CAR modification [149]. These CAR NK cells showed effective cytotoxicity against GPC3-positive tumor cells both in vitro and in vivo and prolonged the survival of CCCO mice without any systemic toxicity and tumorigenicity [149].

It has been reported that anti-folate receptor alpha (αFR) CAR-engineered NK-92 cells specifically target αFR^+^ ovarian cancer (OC) cells [150]. Interestingly, when stimulated with αFR antigen in vitro, the third-generation αFR-specific CAR-modified NK-92 cells displayed not only potent cytokine secretion and degranulation but also high levels of proliferative and anti-apoptotic abilities [150]. Additionally, anti-αFR CAR-engineered NK-92 cells dramatically eliminate OC cells in vivo and effectively delay disease progression in tumor-bearing mice [150]. Mesothelin (MSLN) is another molecule that can be targeted by CAR-modified NK-92 cells in treating OC. MSLN-CAR-engineered NK-92 specifically lysed MSLN^+^ OC cell lines accompanied by high levels of IFN-γ, granzyme B, and GM-CSF. MSLN-CAR NK-92 cells also achieved even more effective tumor control and increased overall survival of NSG mice carrying OC cells, when compared with therapies using CD19-CAR NK-92 or mock control [151]. The researchers further evaluated the feasibility of using MSLN-CAR NK-92 to treat MSLN-positive gastric cancer. They demonstrated that CAR NK-92 cells efficiently infiltrated into the tumor sites and exhibited potent MSLN-dependent cytotoxic activity both in vitro and in the PDX model [151].

A previous study by Hu, Z et al. confirmed that tissue factor (TF) serves as a novel target in 50% to 85% of patients with triple-negative breast cancer (TNBC) [165]. Hu, Z further developed the TF-targeting CAR NK-92MI cell (co-expressing CD16), and a series of preclinical tests were carried out to evaluate its therapeutic effect [153]. He observed that TF-targeting CAR NK cells have direct cytotoxicity against TNBC cells and can mediate stronger ADCC when combined with therapeutic antibodies [153]. Additionally, TF-targeting CAR NK cell therapy also showed potent tumor-directed killing in mouse models of TNBC CDX and PDX [153]. Notably, the epidermal growth factor receptor (EGFR) is another therapeutic target for TNBC. In vitro, EGFR-CAR NK cells exhibited potent cytokine secretion against TNBC cells and mediated specific target cell lysis [154].

Tumor tissues often overexpress NKG2D ligands (MICA/B) that interact with the endogenous activating receptor NKG2D on NK cells. Hence, the adoptive transfer of NKG2D-modified NK cells is an attractive treatment strategy for multiple cancers. A pilot clinical trial study tested the therapeutic effect of NKG2D-modified NK cells in three patients with metastatic colorectal cancer [155]. Researchers confirmed that the extracellular domain of the NKG2D receptor specifically binds to NKG2D ligands expressed on tumor cells and further transmits NK cell activation signals through the intracellular domain of DAP12 [155]. In this study, the number of epithelial cell adhesion molecule (EpCAM)-positive cancer cells in malignant ascites of two patients was dramatically reduced after intraperitoneal infusion of CAR NK cells. Another patient with liver metastases received an ultrasound-guided percutaneous injection of CAR NK cells [155]. A complete metabolic response was detected at the injection site, and the tumor regressed rapidly in the liver region [155]. Notably, CAR modifications were achieved by non-integrating mRNA electroporation technology, and transient expression and multiple infusions ensured the efficacy and safety of clinical application [155]. Patient-derived NKG2D-CAR NK displayed superior cytotoxic activity against MM cells in vitro compared with memory NKG2D-CAR T-cells and mediated more efficient tumor control in a mouse model without any sign of adverse side effects [156]. Notably, advanced tumor cells can utilize the proteolytic shedding of MICA/B to achieve immune escape [166,167,168]. Blocking the shedding of MICA/B [169] or preparing recombinant MICA fusion proteins[170] could enhance NK cell-mediated killing. NKG2D-CAR NK as a combination partner with these immunotherapies may further improve the antitumor efficacy.

A recent study preliminarily evaluated the efficacy and safety of a CAR NK cellular product targeting human epidermal growth factor receptor 2 (HER2) [158]. In terms of product performance, the expression of HER2 CAR further enhanced the specific cytotoxicity of NK cells against various HER2-expressing tumor cell lines [158]. Unlike HER2-specific CAR T, CAR NK elicited more controllable cytokine secretion and had no enhanced cytolysis on non-malignant bronchial epithelial cells expressing basal levels of HER2 [158]. Furthermore, they observed an interesting phenomenon that lentiviral transduction up-regulated the expression of nutrient receptors CD71 and CD98 on the surface of NK cells, leading to a non-specific enhancement of their tumor-killing capacity [158].

##### Challenges to CAR NK Cells Application

Despite the advantages of CAR-NK cells, their clinical application is still limited by several factors, such as insufficient infiltration and homing of immune cells, low persistence in vivo, and immunosuppressive TME, etc.

In anti-tumor treatment, the infiltration of immune cells into tumor sites significantly affects the therapeutic efficacy of CAR NK cells, especially in non-hematological malignancies. To address the issue, modification of chemokine receptors provided a novel strategy to overcome the trafficking barrier and improve the immunotherapeutic efficacy of CAR NK cells against malignancies. Given the high affinity of CXCR1 to IL-8, CXCR1 transduction has been tested to augment the migration and infiltration of NK cells to IL-8-secreting tumor cells [157]. Importantly, compared with NK cells expressing only the NKG2D-specific CAR or mock control, NK cells co-expressing CXCR1 and the NKG2D CAR significantly relieved tumor burden and prolonged survival of mice carrying SKOV3 xenografts [157]. Genetic engineering of CAR NK cells with genes encoding the CXCR4 to increase tumor homing and penetration has also been explored. A recent study demonstrated that CD19-specific CAR NK cells that co-expressed CXCR4 exhibit augmented chemotaxis toward BM while retaining their CD19-specific cytotoxic activity in vitro [144]. Interestingly, it become apparent that the third-generation VSV-G pseudo-typed lentivirus using polybrene as an enhancer was important for high and stable transduction efficiency, enabling CAR expression in more than 80% of primary NK cells [144].

Lacking cytokines support with the limited persistence of CAR NK cells is also a major obstacle to clinical success. Since IL-15 is a critical cytokine required to maintain NK cells’ survival, proliferation, and activation, there has been an ongoing quest to elevate the persistence and anti-tumor activity of NK cell products. In 2018, Katayoun Rezvani’s group generated the fourth-generation anti-CD19 CAR structure [145]. Compared with conventional CAR modification, this novel construct has several improvements. First, IL-15 is expressed to support NK cell proliferation, persistence, and homing in a xenograft NSG mouse model, while retaining the specific cytotoxicity of traditional anti-CD19 CAR constructs against CD19^+^ tumor targets. Second, the addition of incorporating inducible caspase-9 (iC9), which can be stimulated by a small-molecule dimerizer to rapidly eliminate transduced cells, augmented the controllability and safety of adoptive therapy. Third, the source of adoptive cells was optimized by using UCB-derived allogeneic NK cells to guarantee sufficient numbers for ex vivo therapy [145]. Recently, to further boost the anti-tumor effect of IL-15-secreting CAR NK cells, investigators knocked out their *CISH* gene, which encodes the immune checkpoint protein, cytokine-inducible Src homology 2-containing protein (CIS) [131]. They revealed that the removal of CIS in CAR19/IL-15 NK cells completely abolished the inhibition of IL-15 signaling, which in turn activated the Akt/mTORC1-c-MYC signaling axis, thus elevating aerobic glycolysis [131]. In relapsed/refractory B-cell malignancy xenografts model, this upgraded therapy showed potent metabolic fitness and tumor control without any measurable toxicity or malignant transformation [131]. Another study revealed that anti-NKG2D-CAR NK cells also specifically killed AML cells and that ectopic expression of IL-15 could further improve the cytolytic activity and persistence of CAR NK cells [146]. Encouragingly, in a mouse model of KG-1 AML, NKG2D CAR/IL15-NK treatment significantly prolonged the survival of mice and even achieved complete tumor regression after multiple injections [146]. During this study, electroporation of a piggyBac transposon platform was critical for high transduction efficiency, enabling more than 60% of PB-NK cells to stably co-express anti-NKG2D CAR and IL-15 [146]. Moreover, a preclinical study proved that CAR-NK cells targeting prostate stem cell antigen (PSCA) also kept a potent killing effect against pancreatic cancer (PC) cells after 1 cycle of freeze-thaw, which might largely benefit from the co-expression of IL-15 [92]. And this cryopreservation cellular product could survive over 90 days in a metastatic PC model, further supporting the application of NK as an “off-the-shelf” product [92].

Another major challenge in solid cancer immunotherapy is the immunosuppression elicited by intrinsic inhibitory ligands, including PD-L1, B7-H3, and HLA-G. Encouragingly, a novel chimeric PD-1-NKG2D-41BB receptor has been investigated to reverse the immune-suppressive effects of PD-1 [136]. Compared with normal NK-92 cells, CCCR modification markedly augmented the cytotoxicity of NK-92, thereby triggering extensive pyroptosis of PD-L1-positive lung carcinoma H1299 cells in vitro [136]. B7-H3-targeting CAR-engineered NK-92MI cells were also designed to accelerate their degranulation activity and specific cytotoxicity against NSCLC. In the NSCLC xenografts model, the anti-B7-H3 CAR NK-92MI cell further unfolded their excellent immunotherapeutic efficacy. Additionally, anti-B7-H3-CAR NK-92 cells are also served as a promising cellular product for the therapy of melanoma. In either 2- or 3-dimensional assays in vitro, CAR-modified NK-92 cells can overcome the major challenges of TME, including hypoxic conditions, immunosuppressive molecules such as TGF-β, and an acidic pH environment, thus successfully migrating and infiltrating into the tumor site and specifically dissolved MM cell lines [118]. To counteract immune escape in solid tumors, Jan et al. developed a CAR construct targeting HLA-G, consisting of an anti-HLA-G antibody scFv fragment, a KIR2DS4 linkage region, the signaling adaptive molecule DAP12, and the suicide iCasp9 protein [159]. In this study, they further elucidated the anticancer mechanism of this CAR NK strategy. HLA-G CAR competes with inhibitory receptors of NK cells (LILRB1 and KIR2D4) to bind HLA-G on malignant tumors, contributing to the upregulation of phosphor-Syk/Zap70 and the downregulation of phosphor-SHP-1 [159]. Hence, anti-HLA-G CAR NK cells successfully converted immunosuppressive signals into activation signals, triggering HLA-G-depend-cytotoxicity of NK cells in vitro and effectively ablating tumors in orthotopic mice models [159]. Notably, they revealed that low-dose chemotherapy-induced accumulation of HLA-G (involving suppression of DMNT1 and TAP-1 promoter demethylation) further improves the anti-tumor efficacy of HLA-G CAR NK cells [159].

Recently, researchers found that a high concentration of hydrogen peroxide in TME significantly impaired the vitality and cytotoxicity of NK cells, whereas IL-15 could protect NK cells from redox stress induced by H_2_O_2_ via transiently up-regulating the expression of peroxidase-1 (PRDX1) [160]. To compensate for the deficiency of the antioxidant defense system of NK cells, they used a genetic modification strategy to stably over-express PRDX1 in PD-L1-CAR NK-92MI cells [160]. As expected, PRDX1 modification supported the proliferation and survival of NK-92MI under oxidative stress and further improved their anti-tumor cytotoxicity both in vitro and in the breast cancer CDX model [160]. In addition, to resist the immunosuppressive effects mediated by Treg cells in TME, an anti-CD25 CAR-expressing NK-92 cell was designed, which specifically lysed CD25-bearing Jurkat cells with appropriate cytotoxicity and IFN-γ cytokine production [147].

In summary, CAR NK is a novel and promising strategy with great potential to become an “off-the-shelf” allogeneic cell therapy against multiple cancers. To expand the application of CAR NK cells, researchers have designed different antigen receptors for different therapeutic targets to trigger the specific killing of NK cells against corresponding tumor cells. Numerous efforts have been devoted to optimizing the structure of CAR NK to alleviate immunosuppression and enhance their anti-tumor activity, including using advanced genetic engineering techniques to enable NK cells to simultaneously express IL-15 or knock out checkpoint molecules. Additionally, the introduction of the suicide gene further ensures the stability and safety of CAR NK products.

#### 3.2.2. Enhancement of NK Cell Function

In addition to redirected modifications based on CAR structure, other genetic engineering strategies have also been explored to enhance the anti-cancer efficacy of NK cells (Table 2). These strategies mainly focus on two aspects: augmenting NK cell cytotoxicity and improving persistence in vivo (Figure 4).

##### Enhancing Cytotoxicity

Rapid advances in gene-editing technologies have provided multiple strategies for optimizing the cytotoxicity of NK cells, including enhancing ADCC function, inducing target specificity, and releasing immunosuppression.

Monoclonal antibody-based immunotherapies, such as trastuzumab and rituximab, have brought hope to many cancer patients, and ADCC has been reported as one of the major mechanisms affecting its clinical efficacy [26]. As a critical receptor mediating the ADCC process, CD16a crosslinks with the Fc region of IgG Abs on immunoglobulin-opsonized cells to initiate the phosphorylation of intracellular signal adaptor CD3ζ and FcRγ, thereby inducing the activation of NK cells [186,187]. Notably, CD16 expression and function are negatively regulated by recombinant a disintegrin and metalloprotease metalloprotease-17 (ADAM17). It has been shown that ADAM17 inhibits ADCC by rapidly cleaving CD16a when NK cells are stimulated by cytokines or target cells [188,189]. Targeting ADAM17 effectively prevented CD16a shedding of NK cells and significantly enhanced INF-γ secretion [188]. However, given the broad expression of ADAM17 and the numerous hydrolysis substrates, the application of ADAM17 inhibitors lacks specificity and may impair normal biological processes. By contrast, genetic modification of CD16a is a better option to prevent the loss of CD16a on tumor-infiltrating NK cells. The antibody binding affinity of CD16a is susceptible to single nucleotide polymorphism at its amino acid position 158. Specifically, CD16a with valine at the same position (158V) has a higher affinity for mAbs compared with CD16a with phenylalanine at the position 158 [190,191,192]. A recent study demonstrated that the expression of high-affinity CD16 (allotype-V158) receptor confers high-affinity for NK-92 binding to the Fc fragment of therapeutic mAbs, thereby promoting ADCC-mediated lysis of target cells in vitro [122]. Bruce Walcheck’s team used site-directed mutagenesis to replace the proline at position 197 in the middle of the cleavage region with serine and manufactured a non-cleavable version of CD16a (CD16a/S197P) [171]. They did observe that engineered S197P mutation significantly inhibits CD16a receptors shedding from NK-92 cells or iPSC-derived NK cells without disrupting their function [171]. They further introduced a 158V high-affinity variant in the CD16a/S197P structure and successfully engineered iPSC–derived NK cells with high-affinity non-ADAM17 cleavable CD16a (hnCD16-iNK) via a human pluripotent stem cell-like platform [98]. hnCD16-iNK cells could effectively counteract activation-induced cleavage of CD16a to maintain their stable expression [98]. In an ex vivo cytological experiment, hnCD16-iNK cells outperformed PB-NK and unmodified iNK cells for numerous tumor cell-mediated ADCC activities, exhibiting higher levels of cytokine secretion and target cell lysis [98]. In Raji-Luc or SKOV-3-Luc xenograft mouse models, Ab therapy combined with multiple infusions of hnCD16-iNK cells maximized ADCC activity in vivo, contributing to tumor regression and prolonged mouse survival [98]. In a human lymphoma systemic tumor mode, three tumor-bearing mice even achieved complete remission after receiving hnCD16-iNK cells combined with anti-CD20 mAbs and survived for more than 100 days [98]. In addition to direct modification of CD16a, seeking alternative structures of CD16a is also a new direction for investigators. CD64, which also belongs to the Fcγ receptor family, has become an ideal substitute for CD16a for its high IgG affinity and non-cleavable properties. CD64 is primarily expressed on the surface of monocytes, macrophages, or activated neutrophils and initiates the transduction of downstream activation signals through the phosphorylation of ITAM on intracellular FcRγ [193,194]. Studies have shown that CD64-engineered NK-92MI cell lines can also transmit activation signals via CD3ζ to mediate ADCC [173,195]. A novel recombinant receptor, called CD64/16a, has been developed that significantly enhanced the ability of NK-92 and iPSC-derived NK cells to capture soluble tumor-targeting mAbs and to mediate stronger ADCC [172]. The latest study further revealed that intact CD64 could also be stably expressed on NK-92MI cells and induced efficient ADCC. In a prostate cancer xenograft mouse model, adoptive NK-92MI^CD64^ cell transferred with anti-TROP2 and B12 mAb led to excellent tumor control and maintained long-term survival of tumor-bearing mice [173].

To further elevate the specific killing ability of NK cells against malignancies, researchers have carried out a bold innovation of genetically engineering the *TCR* gene for an NK-based therapeutic strategy. A preclinical study endowed NK-92 cells with human CD3 polycistronic complex (CD3γ, δ, ε, ζ dimers) and TCR α/β constructs in two steps by a retrovirus, finally turning NK-92 into a T-cell-like effector cell [174]. TCR-NK-92 exhibited T cell-related phenotype, metabolic and functional characteristics while retaining its NK cell function, and achieved similar tumor control as redirected T cells in tumor-bearing mice [174]. Another study created TCR-NK92 and TCR-YTS with a similar approach, perfectly avoiding mismatches between endogenous and modified TCR subunits [175]. After successful expression of antigen-specific TCR, NK-92 and YTS triggered strong TCR-mediated antigen-specific degranulation and cytokine secretion and showed rapid targeting of tyrosinase-positive melanoma cells, effectively inhibiting tumor growth [175]. Glycoengineering is a simple and efficient approach to the genetic modification of NK cells. Arming NK-92 cells with high-affinity and specific CD22 ligands via glycan metabolic engineering or glycopolymer insertion, rather than CAR engineering, substantially improved their ability to bind and lyse CD22^+^ lymphoma cells in vitro and xenograft model [176]. Recently, Hong and colleagues further induced selectin ligands into CD22-targeting NK-92MI cells via cell-surface chemoenzymatic glycan editing to promote their trafficking to B lymphoma [126].

Many malignancies evade immune control by generating a pro-tumorigenic environment dominated largely by TGF-β [177,196]. In the treatment of glioblastoma multiforme (GBM), disruption of the *TGFBR2* gene in NK cells could restore the antitumor activity of NK cells [177]. Smad3, a key mediator of TGF-β signaling, serves as a precise target for NK cells to attenuate TGF-β-mediated immunosuppression. Indeed, silencing Smad3 restored IFN-γ production by upregulating E4BP4 transcription, thus significantly enhancing NK-92 cell-mediated cytotoxicity against cancer [178]. Interestingly, Burga et al. genetically engineered UCB-NK cells with variant TGF-β receptors, which achieved successful conversion of TGF-β-mediated negative signaling into activating signaling [179]. They revealed that novel TGF-β receptor-modified NK cells blocked the phosphorylation of Smad2 and Smad3, and further promoted the activation of NK to kill neuroblastoma cells in a TGF-β-rich environment [179]. It is widely accepted that the E3 ligase CBLB is a critical negative regulator of lymphocytes. CBLB deletion in placental CD34^+^ cell-derived NK (PNK) cells improved proliferation and augmented cytotoxic effect against multiple tumor cells in vitro [180]. Moreover, CBLB^KO^ PNK cells successfully proliferated and matured in NSG mice and showed improved leukemia control [180]. Notably, the CRISPR/Cas9 technology provides high CBLB editing efficiency in more than 90% of PNK cells, without any impact on cell proliferation, differentiation, or phenotype [180].

Activation and migration of NK cells also play a critical role in exerting their anti-tumor efficacy. Despite the overexpression of DNAM-1 ligand (CD112 or CD155) on human primary sarcoma cells, a preclinical study observed that NK cells minimally infiltrated into sarcomas and tended to downregulate the expression of activating receptors DNAM-1 and NKG2D [181]. This study further confirmed that overexpression of DNAM-1 or NKG2D enhanced the degranulation and cytokine secretion of NK cells, making them more effective in targeting sarcoma explants and multiple tumor cells [181]. Activation with IL-2, cryopreservation, and ex vivo expansion always down-regulate CXCR4 expression on NK cells, further impairing their migration to SDF-1α [182]. Inspiringly, the expression of gain-of-function (GOF) variant CXCR4^R334X^ in PB-NK cells could reverse the reduction of CXCR4 and facilitate chemotaxis to SDF-1α [182]. Furthermore, CXCR4-expressing cells performed superior BM homing ability than control NK cells after adoptive transplantation into NSG mice [182].

##### Improving Persistence

It is essential to maintain the viability and number of NK cells to improve persistence in vivo. Adoptive NK cell immunotherapy often relies on IL-15 or IL-2 to support NK cell viability, but these cytokines may carry serious dose-dependent toxicities. To minimize or avoid the necessity for exogenous cytokines, novel targets of NK cell genetic engineering have been explored recently. For example, the deletion of *CISH* activated IL-15-mediated JAK-STAT signaling and significantly improved iPSC-derived NK in vitro proliferation and in vivo persistence [97]. Activation of IL-15 signaling further promoted mTOR-mediated metabolic fitness, including more efficient glycolysis and oxidative phosphorylation activity, which directly augmented the anti-tumor activity of *CISH*-knockout NK cells under low cytokines concentration [97]. Another study generated hematopoietic growth factor receptors and modified NK cells to replace the administration of exogenous cytokines. Under the stimulation of cognate ligand combined with low-dose IL-2, exogenous expression of EPOR or c-MPL boosted the proliferation and survival of NK-92 in vitro by upregulating Bcl-2 and BclxL [183]. Even in primary human NK cells, modification of c-MPL also significantly improved their proliferation, tumor cytotoxicity, and persistence in vivo [183].

A negative regulating mechanism that hinders NK cell survival is that they lyse their neighboring NK cells expressing the same target via the “missing self” mechanism, which is called fratricide. For NK cell therapy, fratricide always causes a rapid elimination of NK cells, thereby impairing the overall efficacy. As a monoclonal antibody targeting CD38, daratumumab (DARA) has dramatically improved the therapeutic outcome of patients with myeloma [197,198]. However, DARA also induces rapid depletion of CD38^+^ autologous NK cells during treatment, thereby crippling DARA-mediated ADCC [199]. Investigators knocked out CD38 in PB-NK cells using Cas9 ribonucleoprotein complexes, efficiently and precisely [184]. CD38^KO^ cells successfully resisted DARA-induced fratricide with potent ADCC activity and cytotoxicity, indicating a novel strategy to reinforce the therapeutic effect of DARA [184]. A recent study simultaneously endowed CD38^KO^ NK cells with a high-affinity CD16a-158V receptor which further elevated DARA-induced ADCC against myeloma cells [185]. In addition, triple-gene-edited CD38^KO^ iPSC-derived NK cells, co-expressing non-cleavable CD16a and IL-15/IL-15R fusion protein, have achieved desirable results in the tests of their in vivo persistence and tumor-directed cytotoxicity [200]. In clinical practice, another potential risk of adoptive NK cell transfer is the immune rejection of alloreactive host T cells, which limits the persistence of NK cells in vivo and reduces the possibility of multiple infusions. To inhibit the surface expression of HLA class I molecules and prevent autolysis or fratricide of non-HLA-matched NK cells, Hoerster et al. manufactured a double-modified NK cell with *B2M* gene knockout and co-expressed single-chain HLA-E, a ligand for the inhibitory receptor NKG2A [201]. These cells successfully escaped the immune rejection of alloreactive host T-cells while still retaining the original phenotype and cytotoxicity of PB-NK cells [201].

Collectively, genetic engineering confers superior cytotoxicity and persistence to NK cells, laying a solid foundation for the broad clinical application of NK cell-directed therapies.

## 4. Clinical Applications

As an emerging therapeutic strategy, NK ACT has been actively tested in the clinic (Table 3 and Table 4). Current NK-based clinical trials mainly focus on leukemia, especially as consolidation therapy for AML, a common hematopoietic stem cell malignancy with high heterogeneity. In the past few years, several studies have revealed the safety and therapeutic potential of NK cell infusion in the clinical treatment of AML. When co-cultured with radiated K562-mbIL-21-41BBL, NK cells achieved 1024-fold clinical-grade expansion efficiency from PBMCs after 2 weeks of culture [88]. These expanded NK cell products exhibited stronger degranulation and achieved approximately 50% sustained remission after infusion in AML patients with persistent or relapsed minimal residual disease (MRD) [88]. The latest clinical trial indicated that six doses of cryopreserved NK cells in 13 patients with relapsed or refractory AML resulted in an overall response rate of 78.6% and a complete response rate of 50%, with no observed serious adverse events [129]. Adoptive NK therapy elicited beneficial local inflammatory responses or lesion improvement in 4 AML patients with concurrent CNS disease [129]. Recently, Katayoun Rezvani’s group conducted a pioneering clinical trial to evaluate the safety and efficacy of anti-CD19 CAR NK cell products against CD19-positive lymphoid tumors [13]. This phase I/II trial showed that infusion of CAR NK cells induced complete remission in 7 of 11 patients with CD19-positive lymphoid tumors, and all the patients are well tolerable without any dose-limiting toxicity [13]. Although patients received only a single NK cell infusion, the expansion and persistence of CAR NK cells in vivo for at least 12 months can be observed, which may largely benefit from the genetic modification of IL-15 [13]. Researchers always combine adoptive NK therapy with rhIL-2 or rhIL-15 administration to facilitate NK cell activation and expansion in vivo. For the reason of the activation and expansion of Treg cells by low-dose rhIL-2, current clinical trials mainly focus on the rhIL-15 and CAR-NK combination. For example, in a phase I trial of 26 refractory AML patients, combing haploidentical NK cells with rhIL-15 intravenously, 36% achieved potent NK-cell expansion in vivo at day 14 and 35% obtained remission [202]. A retrospective study enrolling 109 chemotherapy-refractory AML patients revealed that infused NK cells maintained well survival and expansion in the BM, and patients with higher NK cell density achieved better leukemia control [203]. Furthermore, pretreatment with CD25-targeting immunotoxin (denileukin diftitox) or low-dose radiation is beneficial for NK cell homing and persistence in the BM [203]. Although several studies have demonstrated the safety and feasibility of infusing KIR-HLA mismatched NK cells as adjuvant therapy in pediatric AML patients, the efficacy of these cell products is not obvious. Possible reasons include the suboptimal number of NK cells infused, limited persistence, and difficulty in patient recruitment. For example, a phase II clinical trial showed that the administration of unmanipulated NK cells did not substantially prolong the event-free survival in children with AML as expected [204]. In another phase II clinical trial, although NK cells were activated and expanded by K562-MB15-41BBL feeder cells before infusion, the efficacy assessment was limited by the number of participants (only 7 patients) [14]. Encouragingly, a prospective phase II clinical trial involving up to 6 institutions is currently being prepared to assess the clinical efficacy of administrating activated and expanded NK cells for AML [205]. Taken together, with proper processing, adoptive NK cell-based immunotherapy is a safe and promising strategy to complement current polychemotherapy-based leukemia treatment options.

Results of clinical trials associated with solid neoplasms have also been reported, including NSCLC [4], primary liver cancer (PLC) [89], GBM[207], medulloblastoma [206], and breast cancer [99]. Interestingly, Multhoff et al. found that NK cells incubated with Hsp70 peptide (TKD) and low-dose IL2 ex vivo could specifically target membrane heat shock protein 70 (mHsp70), an indicator of high-risk tumors (including NSCLC) [208]. This group further confirmed that mHsp70-positive patients with advanced NSCLC receiving 4 cycles of TKD/IL2-activated autologous NK after radio-chemotherapy had improved progression-free survival [4]. Moreover, when combined with PD-1 inhibition, mHsp70-targeting NK cells infusion after radio-chemotherapy effectively induced anti-tumor immunity and achieved long-term tumor control in an unresectable NSCLC patient [209]. In a small-scale preliminary phase II trial, allogenic NK infusion combined with irreversible electroporation (IRE) therapy presented a synergistic effect in PLC, significantly reducing circulating tumor cells and extending patients’ median progression-free survival (15.1 months) and 1-year overall survival (23.2 months) [89]. Compared with intravenous infusion, locoregional infusion of NK cells appears to be more beneficial in achieving immune infiltration of solid neoplasms. For example, early results of a phase I study supported that intraventricular infusion of allogeneic NK cells in children with recurrent medulloblastoma and ependymoma was safe and feasible and the cryopreservation did not impair the activity and function of NK cells [206]. In addition, the safety and toxicity of intraperitoneal infusion of NK cells for the treatment of recurrent OC are also under evaluation [210]. Recently, Lee et al. showed a promising result of the combination of ex vivo expanded NK cells with trastuzumab in treating HER2-positive solid malignancies [99]. In this study, all patients are well tolerated without any severe toxicity, and 6 of 19 patients had stable disease lasting longer than 6 months [99]. Notably, one patient with a high-affinity CD16 variant (allotype v158) achieved a partial response, indicating that CD16 genotyping might be a potential biomarker for response to a therapeutic antibody [99].

Collectively, clinical anti-tumor regimens commonly choose NK cell immunotherapy as a post-remission consolidation strategy or in combination with other treatments. Most clinical trials based on NK infusion are in the early exploratory phase with a short follow-up time and a small number of enrolled patients. Although current data indicate that the safety of treatment with NK cells is better than that of T-cells, its clinical efficacy still warrants investigation in larger-scale clinical trials.

## 5. Conclusions and Future Direction

NK cells have extensive tumor-killing effects independent of MHC restriction. The variety of cell sources and recent preclinical advances in genetic engineering have further optimized the potential of NK cells in the ACT. Here, we summarized recent development in the field of NK cell-based adoptive transfer with an emphasis on the source and genetic modification of NK cells. We represented various sources of NK cells currently available for adoptive infusion, ranging from traditional PB-NK cells to emerging ML-NK cells, which have their characteristics to meet different clinical needs. Moreover, we described in detail diverse genetic modification strategies to improve the anti-tumor activity of NK cells, including CAR engineering for different targets, gene mutation or recombination to enhance cytotoxicity, knockout of immunosuppression-related genes, cytokine co-expression to improve migration and persistence in vivo.

However, most NK cell products are still in the early preclinical exploration stage and lack large cohorts of patients to support their clinical efficacy. Moreover, ex vivo expansion and gene transduction of primary NK cells often require high technical requirements because of their demanding culture conditions and intracellular antiviral defense mechanisms [211]. NK-92 cell line has been widely used in preclinical studies based on genetic engineering platforms for its indefinite expansion. However, due to safety concerns, the current clinical trials are mainly based on the infusion of autologous or allogeneic primary NK cells, which raised the expense.

The future direction of adoptive NK cell therapy is to continuously optimize the property of cellular products, making it a safe and effective adjuvant therapeutic strategy. Researchers are exploring multiple advanced genetic modification strategies to maximize the potential of NK cells for the treatment of multiple cancers, especially solid tumors. Meanwhile, it is also important to reduce costs and standardize the procedure of NK preparation and evaluate key preclinical discoveries in larger-scale clinical trials.

## Figures and Tables

**Figure 1 cancers-14-05657-f001:**
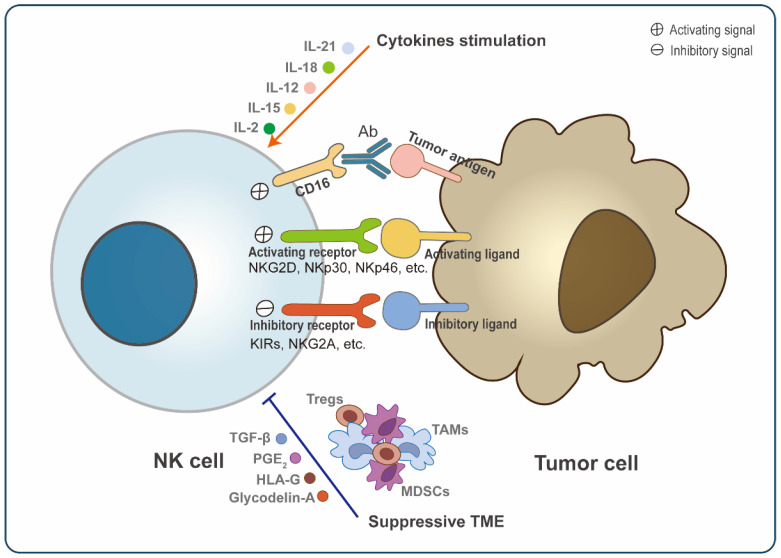
General regulation of NK cell activation. Natural killer (NK) cell activation is a dynamic process mediated by multiple factors. NK cells express a range of receptors, including activating receptors (NKG2D, NKp30, NKp46, etc.) and inhibitory receptors (KIRs, NKG2A, etc.), which bind to corresponding tumor cell ligands and then transmit signals. Moreover, the Fc receptor (CD16) can also transmit activating signals into NK cells via binding with antibody-coated target cells. The stimulation of cytokines, such as IL-2, IL-15, IL-12, IL-18, and IL-21 also promotes the activation and proliferation of NK cells. However, many components in TME can suppress NK cell activation, including soluble suppressive molecules (TGF-β, prostaglandin E2, HLA-G, and glycodelin-A) and immunosuppressive cells, such as regulatory T-cells (Tregs), tumor-associated macrophages (TAMs), and myeloid-derived suppressor cells (MDSCs). Hence, the activation of NK cells ultimately depends on the sum of these signaling. When the activation signal is stronger than the inhibitory signal, NK cells will be activated and kill tumors.

**Figure 2 cancers-14-05657-f002:**
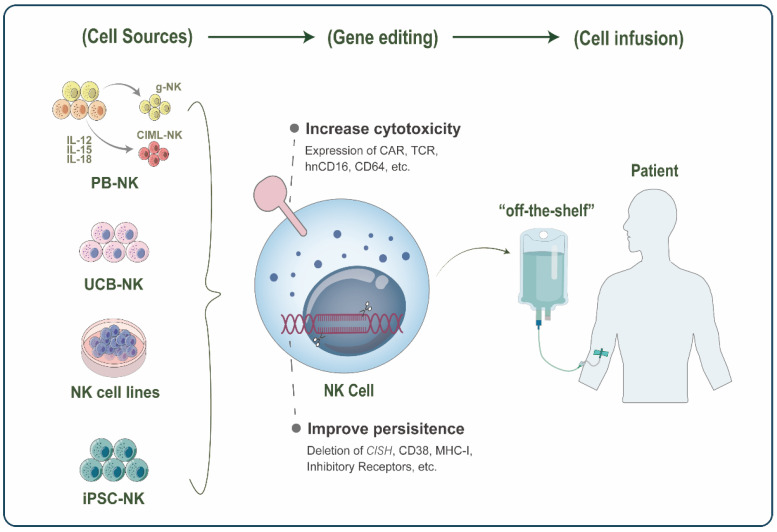
Schematic representation of Natural killer (NK)-based adoptive cell therapy (ACT). NK cells for adoptive infusion come from a wide range of sources, including peripheral blood NK (PB-NK) cells, umbilical cord blood NK (UCB-NK) cells, NK cell lines, and induced pluripotent stem cell (iPSC)-derived NK cells. In PB, the novel FcRg-deficient NK(g-NK) subset and cytokine-induced memory-like NK (CIML-NK) have become promising cell sources in ACT with their enhanced antitumor activity and memory properties. Various gene editing strategies further enhance the function of therapeutic NK cells, with two main focuses: increasing NK cell cytotoxicity and improving persistence in vivo. Finally, these engineered NK cells can be infused into patients as “off-the-shelf” products for cancer therapy.

**Figure 3 cancers-14-05657-f003:**
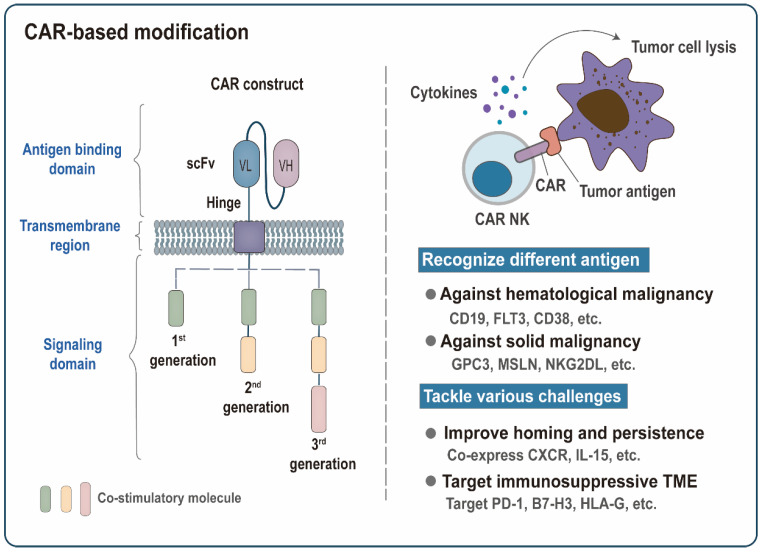
Chimeric antigen receptors (CAR)-based modification. Functional CAR molecule often consists of an antigen-binding domain, a transmembrane region, and an intracellular signaling domain. The extracellular domain contains a single-chain variable fragment (scFv) that binds to tumor antigens. Conventional CAR receptors are divided into three generations. Based on the structure of the first-generation CAR, the second and third-generation CARs with enhanced proliferation and killing capability are formed by adding the intracellular part of one or two costimulatory molecules. CAR molecules confer NK cells the ability to recognize tumor antigens specifically and transmit activation signals into NK cells, thereby triggering the lysis of tumor cells. The development of CAR modification has two main focal points: recognizing different tumor antigens and tackling various challenges. The scFv structures targeting different tumor antigens endow CAK-NK cells with different specificities against multiple malignancies. Co-expression of the chemokine receptors (CXCR) or IL-15 can further promote homing of CAR NK cells to tumor sites or improve their in vivo persistence. Targeting inhibitory receptors PD-1 or inhibitory ligands (B7-H3 and HLA-G) provides an effective way to overcome the immunosuppressive tumor microenvironment (TME).

**Figure 4 cancers-14-05657-f004:**
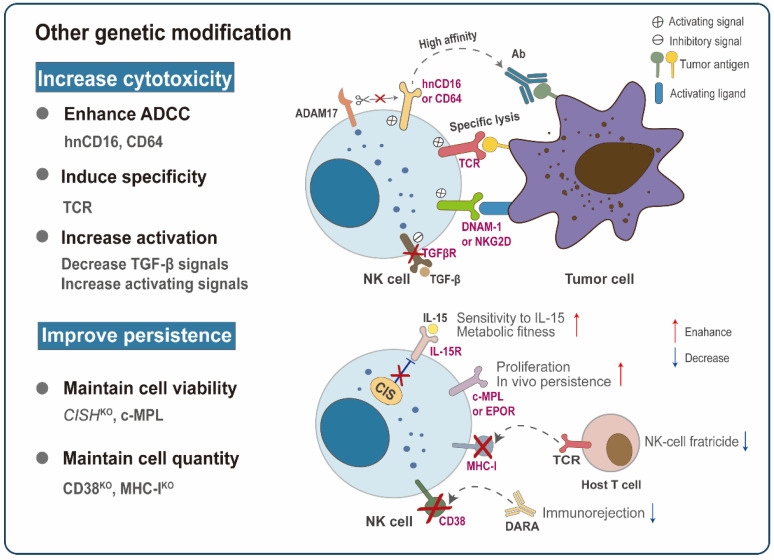
Other advanced genetic modification strategies. Other non-CAR structure-based genetic modifications have also been explored to improve NK cytotoxicity and in vivo persistence. For example, the introduction of high-affinity non-ADAM17 cleavable CD16 (hnCD16) or CD64 significantly enhances the affinity of NK cells to the tumor-targeting antibodies (Abs), thereby improving antibody-dependent cell-mediated cytotoxicity (ADCC). TCR NK cells achieve similar tumor control as redirected T-cells while retaining their NK cell function. Targeting TGF-β signaling or transducing activating receptors (DNAM-1, NKG2D) can tilt NK cells toward activation. Strategies to improve in vivo persistence mainly focus on maintaining the viability and quantity of NK cells. Under low cytokine concentration, possible methods to maintain cell vitality include the deletion of *CISH* or the expression of c-MPL or EPOR. Prolonged persistence can also be achieved by deleting the endogenous CD38 or MHC-I of NK cells so that the infused NK cells are undetectable to the daratumumab (DARA) or the recipient’s T-lymphocytes.

**Table 1 cancers-14-05657-t001:** Current CAR NK cell products.

Cell Source	Target	Generation	Activation Signal	Transduction	Model	Malignancy	Ref.
NK-92	FLT3	Second	CD28-CD3ζ	LV	B-ALL CDX model in NOD-SCID IL2R γnull mice	B-ALL	[22]
NK-92	CD38	Third	CD28-4-1BB-CD3ζ	RV	None	MM, BL	[135]
KHYG1, PB-NK	CD38	Second	CD28-CD3ζ	Electroporation	None	AML	[140]
NK-92	CD5	Second	4-1BB-CD3ζ, 2B4-CD3ζ	LV	CD5^+^T-ALL CDX model in NSG mice	T-ALL	[137]
NK-92MI	CD7	Third	CD28-4-1BB-CD3ζ	Electroporation	T-ALL PDX model in NSG mice	T-ALL	[96]
PB-NK	CD33	Second	4-1BB-CD3ζ	LV	AML CDX model in NSG mice	AML	[141]
NK-92	CD123	Third	CD28-4-1BB-CD3ζ	RV	AML PDX model in NSG mice	AML	[142]
PB-NK	CD123	Second	2B4-CD3ζ	RV	AML CDX model in NSG mice	AML	[143]
PB-NK	CD19, SDF-1	Second	4-1BB-CD3ζ	LV	None	B-cell malignancies	[144]
CB-NK	CD19	Second	CD28-CD3ζ	RV	BL CDX model in NSG mice	CD19^+^leukemia/lymphoma	[145]
CB-NK	CD19	Second	CD28-CD3ζ	RV	BL CDX model in NSG mice	CD19^+^leukemia/lymphoma	[131]
PB-NK	NKG2DL	Third	CD28-4-1BB-CD3ζ	Electroporation	AML CDX model in NSG mice	AML	[146]
NK-92	CD25	Third	CD28-4-1BB-CD3ζ	LV	None	CD25^+^T-ALL	[147]
NK-92, PB-NK	GPC3	Second	CD28-CD3ζ	LV	HCC orthotopic CDX model in NOD/SCID mice	HCC	[148]
NK-92	GPC3	Third	DNAM1-2B4-CD3ζ	LV	None	HCC	[138]
iPSC-NK	GPC3	Third	CD28-4-1BB-CD3ζ	LV	GPC3^+^OC CDX model in NSG/NOG mice	OC	[149]
NK-92	αFR	Three	CD3ζ, CD28-CD3ζ, CD28-CD137-CD3ζ	LV	OC CDX model in B-NDG mice	OC	[150]
NK-92	MSLN	Third	CD28-4-1BB-CD3ζ	LV	OC CDX model in NSG mice	OC	[151]
NK-92	MSLN	Second	2B4-CD3ζ	LV	GC CDX and PDX model in NSG mice	GC	[152]
NK-92MI	TF	Third	CD28-4-1BB-CD3ζ	LV	TNBC CDX and PDX model in NSG mice	TNBC	[153]
PB-NK	EGFR	Third	CD28-4-1BB-CD3ζ	LV	TNBC CDX and PDX model in female nude mice	TNBC	[154]
PB-NK	NKG2DL	First	DAP-12	Electroporation	CRC CDX model in NSG mice; Three patients with chemotherapy-refractory metastatic CRC	CRC	[155]
PB-NK	NKG2DL	Second	4-1BB-CD3ζ	LV	MM CDX model in NSG mice	MM	[156]
PB-NK	NKG2DL, IL-8	First	CD3ζ	Electroporation	OC and HNC CDX model in NSG mice	OC, HNC	[157]
PB-NK	HER2	Second	CD28-CD3ζ	LV	None	BC	[158]
NK-92	PDL1	First	4-1BB	LV	Lung cancer CDX model in NOG mice	Lung cancer	[136]
NK-92MI	B7-H3	Second	4-1BB-CD3ζ	LV	NSCLC CDX model in NOD/SCID mice	NSCLC	[125]
NK-92	B7-H3	Second	CD28-CD3ζ	LV	None	Melanoma	[118]
PB-NK	HLA-G	First	DAP-12	LV	MDA-MB-231 and U87 orthotopic xenograft model in NSG mice	Solid tumors	[159]
NK-92MI	PD-L1	Second	CD28-CD3ζ	LV, Electroporation	BC CDX model in female WT Balb/c, C57BL/6, and NSG mice	BC	[160]

PB, peripheral blood; CB, cord blood; iPSC, induced pluripotent stem cell; LV, lentivirus; RV, retrovirus; CDX, cell-derived xenograft; PDX, patient-derived xenograft; ILC, Innate lymphoid cells; B-ALL, B-cell acute lymphoblastic leukemia; MM, multiple myeloma; BL, Burkitt’s lymphoma; AML, acute myeloid leukemia; T-ALL, T cell acute lymphoblastic leukemia; HCC, hepatocellular carcinoma; OC, ovarian cancer; HNC, head and neck cancer; BC, breast cancer; GC, gastric cancer; TNBC, triple-negative breast cancer; CRC, colorectal cancer; NSCLC, non-small cell lung cancer.

**Table 2 cancers-14-05657-t002:** Current genetic engineering strategies aimed at improving the functions of NK cells.

Cell Source	Modifications	Transduction	Model	Malignancy	Ref.
NK-92	CD16(158V): CD16(allotype-V158)-CD8-4-1BB-CD3ζ	LV	None	Multiple cancers	[122]
NK-92, iPSC-NK	CD16a/S197P: convert the serine at position 197 to a proline in CD16	RV, SB transposon	None	-	[171]
iPSC-NK	hnCD16: high-affinity CD16a 158V variant	SB transposon	B-cell lymphoma/OC CDX model in NSG mice	Multiple cancers	[98]
NK-92, iPSC-NK	CD64/16a: CD64 ECD-CD16(TM + ICD)	RV, SB transposon	None	-	[172]
NK-92MI	CD64	RV	mCRPC CDX model in NSG mice	mCRPC	[173]
NK-92	TCR: hCD3(CD3γ/δ/ε/ζ dimers)-TCR (TCRα/β)	RV	CRC CDX model in NSG mice	CRC	[174]
NK-92, YTS	TCR: hCD3(CD3γ/δ/ε/ζ dimers)-TCR (TCRα/β)	LV	Melanoma CDX model in NOD/SCID mice	Melanoma	[175]
NK-92	CD22 ligand	Glycoengineering	B cell lymphoma CDX model in Balb/c nude mice	B cell lymphoma	[176]
NK-92MI	CD22 ligand and selectins	Glycoengineering	B cell lymphoma CDX model in NSG mice	B cell lymphoma	[126]
PB-NK	*TGFBR2* knockout	CRISPR/Cas9	GBM PDX model in NSG mice	GBM	[177]
NK-92	SMAD3-silencing	LV	Hepatoma and melanoma CDX model in NOD/SCID mice	Hepatoma and melanoma	[178]
CB-NK	Variant TGF-β receptors: (RBDNR: TGF-βRII-truncated CD19-Puro; NKA: TGF-βRII-DAP12-truncated CD19-Puro; NKCT: TGF-βRII-synNotch-RELA-truncated CD19-Puro)	RV	Neuroblastoma CDX model in NSG mice	Neuroblastoma	[179]
PB-NK	CBLB knockout	CRISPR/Cas9	AML CDX model in NSG mice	-	[180]
NK-92	DNAM-1 and NKG2D	LV	None	Sarcomas	[181]
PB-NK	Gain-of-function variant CXCR4R334X	Electroporation	NSG mice	-	[182]
iPSC-NK	*CISH* knockout	CRISPR/Cas9	AML CDX model in NSG mice	AML	[97]
NK-92, PB-NK	EPOR or c-MPL	LV	NSG mice	-	[183]
PB-NK	CD38 knockout	CRISPR/Cas9, Electroporation	NSG mice pretreated with DARA	MM	[184]
PB-NK	CD38 knockout and CD16 (158V)	CRISPR/Cas9, Electroporation	MM CDX model in NSG mice	MM	[185]

PB, peripheral blood; CB, cord blood; iPSC, induced pluripotent stem cell; ECD, extracellular domain; TM, transmembrane; ICD, intracellular domain; TCR, T cell receptor; LV, lentivirus; RV, retrovirus; SB, sleeping beauty; mCRPC, metastatic castration resistant prostate cancer; OC, ovarian cancer; CRC, colorectal cancer; GBM, glioblastoma multiforme; AML, acute myeloid leukemia; MM, multiple myeloma.

**Table 3 cancers-14-05657-t003:** Recent clinical results of NK cell infusions (2019–2021).

Cell Source	NK Cell Preparation	Interventions	Malignancy	Response	Phase	Patient Number	Patient Age	Trial Identifiers
Haploidentical NK	K562-mb21-41BBL	Before treatment with chemotherapy	AML	MRD remission in 9 patients	-	20	≥18	- [88]
Haploidentical NK	K562-mb21-41BBL	Before treatment with chemotherapy	Relapsed/Refractory AML	78.6% overall response; 50.0% CR; CNS responses in 4 patients	I/II	13	2–59	NCT02809092 [129]
UCB-NK	K562-mb21-41BBL; Modification: anti-CD19 CAR, IL-15, iC9	Before treatment with lymphodepleting chemotherapy	CD19^+^lymphoma	CR in 7 patients; Richter’s transformation remission in 1 patient	I/II	11	47–70	NCT03056339 [13]
Haploidentical NK	None	Before treatment with lymphodepleting chemotherapy; After treatment with rhIL-15 intravenously (0.3–1.0 mg/kg)	Refractory AML	Robust NK expansion in 36% of patients at day 14; CR in 32% of patients	I	26	≥18	NCT01385423 [202]
Haploidentical NK	None	Before treatment with lymphodepleting chemotherapy and rhIL-2 subcutaneously	Pediatric AML	None	II	21	0–15	NCT00703820 [204]
Haploidentical NK	K562-mb15-41BBL	Before treatment with lymphodepleting chemotherapy and rhIL-2 subcutaneously	Pediatric AML	CR in 6 patients	II	7	0–15	NCT02763475 [14]
Autologous NK	TKD/IL2-activated	Before treatment with RCT	NSCLC	RCT + NK group (*n* = 8): 67% 1-year probabilities for PFS	II	16	56–76	- [4]
Autologous NK	NK in vitro preparation kit	Before treatment with IRE	PLC	IRE + NK group (*n* = 18): median PFS (15.1 months), median 1-year OS (23.2 months)	II	40	20–80	NCT03008343 [89]
Autologous NK	K562-mb21-41BBL	None	Recurrent pediatric MB and ependymoma	Progressive disease in 9 patients; transient radiographic response in 1 patient	I	9	8–18	- [206]

UCB, umbilical cord blood; AML, acute myeloid leukemia; GBM, glioblastoma multiforme; NSCLC, non-small cell lung cancer; PLC, primary liver cancer; MB, medulloblastoma; RCT, radiochemotherapy; IRE, irreversible electroporation; MRD, minimal residual disease; CR, complete remission; PFS, progression-free survival; OS, overall survival; CNS, central nervous system; CRS, cytokine release syndrome.

**Table 4 cancers-14-05657-t004:** Clinical trials of NK cells for cancer immunotherapy listed on https://www.clinicaltrials.gov/ (1 November 2019–1 November 2022, accessed on 11 November 2022).

NCT Number	Infused Cells	Diseases	Phases	Enrollment	Status
NCT03937895	Allogeneic NK Cells	Biliary Tract Cancer	I/II	40	Completed
NCT04616209	Allogeneic NK Cells	Non-small Cell Lung Cancer	I/II	24	Recruiting
NCT04847466	Irradiated PD-L1 CAR-NK Cells	Recurrent/Metastatic Gastric or Head and Neck Cancer	II	55	Recruiting
NCT05213195	NKG2D CAR-NK Cells	Refractory Metastatic Colorectal Cancer	I	38	Recruiting
NCT03329664	Cytokine-induced Killer (CIK) Cells	Colon Cancer Stage IV	I/II	20	Recruiting
NCT04872634	NK01 (Super Natural Killer Cells 01)	Non-small Cell Lung Cancer	I/II	24	Recruiting
NCT05194709	Anti-5T4 CAR-NK Cells	Advanced Solid Tumors	I (early)	40	Recruiting
NCT04290546	cytokine-induced memory-like NK (CIML-NK) cells	Squamous Cell Carcinoma of the Head and Neck	I	12	Recruiting
NCT05020678	NKX019	B-cell Malignancies	I	60	Recruiting
NCT04796675	Anti-CD19 CAR-NK Cells	B Lymphoid Malignancies	I	27	Recruiting
NCT05099549	SNK01	Advanced/Metastatic EGFR-Expressing Cancers	I/II	121	Recruiting
NCT04143711	DF1001	Solid Tumor, Adult	I/II	220	Recruiting
NCT04319757	ACE1702	HER2-expressing Solid Tumors	I	36	Recruiting
NCT04310592	CYNK-001	Acute Myeloid Leukemia, Adult	I	94	Recruiting
NCT05069935	FT538	Solid Tumor, Adult	I	189	Recruiting
NCT04634435	CIML NK Cells	Multiple Myeloma	I/II	25	Recruiting
NCT05304754	Alloreactive NK cells	High-risk Leukemias	I/II	18	Recruiting
NCT05008536	Anti-BCMA CAR-NK Cells	Multiple Myeloma, Refractory	I (early)	27	Recruiting
NCT04623944	NKX101-CAR NK Cells	Acute Myeloid Leukemia, Adult	I	90	Recruiting
NCT04558931	Autologous NK Cells	Multiple Myeloma	II	60	Recruiting
NCT04901416	Allogeneic NK Cells	Acute Myeloid Leukemia, Adult	I	18	Recruiting
NCT04309084	CYNK-001	Multiple Myeloma	I	29	Active, not recruiting
NCT04630769	FT516	Ovarian Cancer	I	3	Completed
NCT05379647	NK Cells	B-Cell Malignancies	I	24	Recruiting
NCT04259450	AFM24	Advanced Solid Tumor	I/II	155	Recruiting
NCT03821519	Cytokine-induced Killer (CIK) Cells	Relapsed Hematologic Malignancy	I/II	20	Recruiting
NCT04347616	UCB-NK Cells	Acute Myeloid Leukemia, Adult	I/II	23	Recruiting
NCT05247957	NKG2D CAR-NK Cells	Safety and Efficacy	I	9	Recruiting
NCT05008575	Anti-CD33 CAR-NK Cells	Acute Myeloid Leukemia	I	27	Recruiting
NCT05215015	Anti-CD33/CLL1 CAR-NK Cells	Acute Myeloid Leukemia	I (early)	18	Recruiting
NCT04220684	Haploidentical NK Cells	Acute Myeloid Leukemia	I	30	Recruiting
NCT03539406	UCB-NK Cells	Recurrent Ovarian Carcinoma	I	12	Recruiting
NCT05333705	Allogeneic NK Cells	Acute Myeloid Leukemia	I	15	Recruiting
NCT04836390	Haploidentical NK Cells	Acute Myeloid Leukemia	II	30	Enrolling by invitation
NCT04898543	M-CENK	Metastatic Solid Tumor	I	30	Recruiting
NCT04551885	FT516	Solid Tumor, Adult	I	12	Active, not recruiting
NCT05182073	FT576	Multiple Myeloma	I	168	Recruiting
NCT03348033	Autologous NK Cells	Chronic Myeloid Leukemia	I/II	5	Enrolling by invitation
NCT04614636	FT538	Advanced Hematologic Malignancies	I	105	Recruiting
NCT04162158	Allogeneic NK Cells	Hepatocellular Carcinoma	I/II	200	Recruiting
NCT05108012	NK Cells	Glioblastoma Multiform	I	5	Recruiting
NCT02573896	NK Cells	Neuroblastoma	I	13	Active, not recruiting
NCT04802070	Autologous CIK Cells	Sarcoma	I	36	Recruiting
NCT04887012	Anti-CD19 CAR-NK Cells	B-cell Non-Hodgkin Lymphoma	I	25	Recruiting
NCT04489420	CYNK-001	Glioblastoma	I	3	Terminated
NCT04074746	AFM13-NK	Recurrent/Refractory CD30 Positive Lymphomas	I/II	30	Recruiting
NCT05137275	Anti-5T4 CAR-raNK Cells	Locally Advanced or Metastatic Solid Tumors	I (early)	56	Recruiting

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
