# Peer review of "Natural Killer Cells: A Promising Kit in the Adoptive Cell Therapy Toolbox"

_cancers, 2022, doi:10.3390/cancers14225657_

Round 1
Reviewer 1 Report
Authors provided a very exhaustive review on the approaches using NK cell for cancer immunotherapy. The text is well presented, sections properly articulated. Figures are clear and exhaustive.
I have minor comments/suggestions that I feel it would improve the quality of the review.
Authors described the anti-tumor functions of NK cells but totally missed to discuss pro-tumor functions of angiogenesis, such as pro-metastatic features (10.1016/j.ccell.2017.06.009 ; 10.1038/s43018-020-0081-z) and pro-angiogenic activities (10.3390/cells10071621; 10.3390/cancers11040461). Also, authors did not mention data on the capability of NK cells to express markers of cell exhaustion, such as PD-1, CD96, Tigit, TIM-3.
These points should be cited and discussed, since NK cells can be turned from tumor enemies to tumor allies, due to their cell plasticity/polarization state and based on the tumor microenvironment cell (MDSCs, Treg, TAMs) and soluble factor (TGFb, Adenosine, LDH and acidity, soluble HLAg, exosomes, miRs, hypoxia) interactions. These interactions could compromise the activity of NK cells armed for immunotherapy approaches.
I strongly suggest authors to provide a table showing the clinical trials (at any stage) employing NK cells for cancer immunotherapy. This would significantly complete the overall view of the specific topic.
Lines 39-40: “Hence, T or NK-based adoptive transfer strategy has provided new hope for patients with advanced malignant tumors [1]”: I think that more than one reference should be provided. Please, also anticipate in the text which “advanced malignant tumors” gains from “T or NK-based adoptive transfer strategy”
Lines 49-50: “enhance the adaptive immune function by secreting various cytokines”. Please go into details on “adaptive immune functions” and indicate specifically which of the “various cytokines” are secreted.
Line 83: once again, specify which cytokines
Line 84: What does “quantitative advantage” mean? Please rephrase.
Line 96: explain CD16 at first appearance
Author Response
Dear Reviewer:
Thank you for your comments concerning our manuscript entitled “Natural Killer Cells: A Promising Kit in the Adoptive Cell Therapy Toolbox”. (ID: cancers-1986173). Those comments are all valuable and very helpful for revising and improving our paper. We have studied the comments carefully and have made corrections which we hope meet with approval. We have provided a point-by-point response to your comments. Please see the attachment.
Response to Reviewer 1 Comments
Reviewer #1: Authors provided a very exhaustive review on the approaches using NK cell for cancer immunotherapy. The text is well presented, sections properly articulated. Figures are clear and exhaustive.
Reply: Thank you very much for reviewing our manuscript and giving your valuable comments.
Point 1: Authors described the anti-tumor functions of NK cells but totally missed to discuss pro-tumor functions of angiogenesis, such as pro-metastatic features (10.1016/j.ccell.2017.06.009 ; 10.1038/s43018-020-0081-z) and pro-angiogenic activities (10.3390/cells10071621; 10.3390/cancers11040461). Also, authors did not mention data on the capability of NK cells to express markers of cell exhaustion, such as PD-1, CD96, Tigit, TIM-3.
These points should be cited and discussed, since NK cells can be turned from tumor enemies to tumor allies, due to their cell plasticity/polarization state and based on the tumor microenvironment cell (MDSCs, Treg, TAMs) and soluble factor (TGFb, Adenosine, LDH and acidity, soluble HLAg, exosomes, miRs, hypoxia) interactions. These interactions could compromise the activity of NK cells armed for immunotherapy approaches.
Response 1: Thank you for your advice. We are very sorry for our negligence of these important points. According to your suggestion, we described some pro-tumor functions of NK cells, including promoting angiogenesis and metastasis as well as the expression of depletion markers of NK cells. The content on NK cell plasticity/polarization and interaction with TME-related cell components and soluble factors has been supplemented. Please see the revised text from lines 156 to 187. Related references were also cited.
Point 2: I strongly suggest authors to provide a table showing the clinical trials (at any stage) employing NK cells for cancer immunotherapy. This would significantly complete the overall view of the specific topic.
Response 2: Thank you for your advice. We have overviewed CAR-NK cell clinical trials worldwide in last 3 years listed on https://www.clinicaltrials.gov/. (Please see Tabe 4).
Point 3: Lines 39-40: “Hence, T or NK-based adoptive transfer strategy has provided new hope for patients with advanced malignant tumors [1]”: I think that more than one reference should be provided. Please, also anticipate in the text which “advanced malignant tumors” gains from “T or NK-based adoptive transfer strategy”
Response 3: Thank you for your advice. We cited 5 new references based on T or NK-based-adoptive transfer strategy to support our view and point out the specific types of advanced cancer. Please see the revised text from lines 43 to 46.
Point 4: Lines 49-50: “enhance the adaptive immune function by secreting various cytokines”. Please go into details on “adaptive immune functions” and indicate specifically which of the “various cytokines” are secreted.
Response 4: Thank you for your advice. We have made corresponding modifications according to your suggestions: we specifically indicated cytokines secreted by NK cells (including IFN-γ, TNF, XCL1, CCL3, CCL4 CCL5, IL-13, and GM-CSF); we pointed out that NK cells indirectly enhance adaptive immunity by secreting IFN-γ, which induces Th1 polarization and activate CD8+ T cells. And we moved this content to 2.2.1. Anti-tumor function, hoping to make our manuscript more fluent. Please see the revised text from lines 150 to 155.
Point 5: Line 83: once again, specify which cytokines
Response 5: Thank you for your advice. We added the specific name of cytokines secreted by CD56bright CD16- NK cells (IFN-γ, TNF, IL-10, IL-13, and GM-CSF). Please see the revised text from lines 94 to 95.
Point 6: Line 84: What does “quantitative advantage” mean? Please rephrase.
Response 6: Thank you for pointing out this problem. “the CD56dimCD16+ subset has a quantitative advantage in peripheral blood (PB)” has been modified to “the CD56dimCD16+ subset accounts for about 90% of all NK cells in peripheral blood (PB) ” Please see the revised text from line 96.
Point 7: Line 96: explain CD16 at first appearance
Response 7: Thanks for reminding us. We have added corresponding explanations. Please see the revised text from line 106.

Reviewer 2 Report
Xiao and the authors reviewed the utilization of natural killer cells as a promising kit in adoptive cell therapy. They gave an overview of NK cells, NK-based adoptive cell therapy with various cell types such as PB-NK, UCB-NK, cell lines and stem cell-derived NK. Further, CAR-modified NK cells, then CAR-NK on haematological malignancy and solid malignancy. Finally discussed the Challenges in CAR NK cells application and then, finally summarized the Clinical applications, conclusion and future direction. This is a comprehensive and well-written manuscript with figures and tables.
I recommend the publication of the manuscript in its current form.
Author Response
Dear Reviewer:
Thank you very much for reviewing our manuscript entitled “Natural Killer Cells: A Promising Kit in the Adoptive Cell Therapy Toolbox” (ID: cancers-1986173) and giving your valuable comments. Thank you again and best regards.
Reviewer 3 Report
Natural Killer Cells: a promising Kit in the adoptive cell therapy Toolbox
The review article from Zoe and collaborators is very comprehensive and carefully evaluates the role of NK cells in ACT.
I have some comments that I hope will make the review better.
The subject is complex and there are certain concepts that are mentioned and not explained. I assume that there is the expectation that the readers will know them, but if they are reading a review they may well not know.
For example:
‘off-the-shelf’, please explain better (as you do in one of the last paragraphs, lane 663) why NK can be used more securely that T cells (alloimmunity, rejection, cost, expandability…), and so became a ‘of-the –shelf’. What other characteristics are important for this cells to be ‘of-the shelf’?
Mention the benefits of the suicide gene- lane 286
What does iCasp9 means in lane 308?
Mention what nanobodies are- lane312
What are the 4 different components? Lane 343
Explain what a master cell bank is- homo-haplotype?- lane 366
The figure shown is very comprehensive, in a way that may be too mcu information in one figure, making part B and C very confusing, as there is too much information that does not clearly matches the figure, for example, what is the triangle is meant to mean in 1B? decrease of something? 1B and 1 C, can be simplify in this figure (basic one), and made 2 more figures that show more information on B and C.
A figure of the general regulation of NKs activation, it will be very useful, and adding the modifications done on it.
#
When speaking about different experimental work, the complexity of the models is great, so describing them is complex. The authors added almost in all cases how (methodology) the cells were generated (lentiviral, retoviral and so on). This information is not vital for the message, complicates the sentences and is already present in the Tables.
Table 1, can be divided into solid tumors vs hematological ones
Typos/wording
Lane: 258 typically applied , will be best used
When mentioning a gene or gene product, please use scientific nomenclature CISH no CISH
Disolved ? lane 496
Even lane 579 is not needed
Author Response
Dear Reviewer:
Thank you for your comments concerning our manuscript entitled “Natural Killer Cells: A Promising Kit in the Adoptive Cell Therapy Toolbox” (ID: cancers-1986173). Those comments are all valuable and very helpful for revising and improving our paper. We have studied the comments carefully and have made corrections which we hope meet with approval. We have provided a point-by-point response to your comments. Please see the attachment.
Response to Reviewer 3 Comments
Reviewer #3: The review article from Zoe and collaborators is very comprehensive and carefully evaluates the role of NK cells in ACT.
Reply: Thank you very much for reviewing our manuscript and giving your valuable comments.
Point 1: ‘off-the-shelf’, please explain better (as you do in one of the last paragraphs, lane 663) why NK can be used more securely that T cells (alloimmunity, rejection, cost, expandability…), and so became a ‘off-the-shelf’. What other characteristics are important for this cells to be ‘off-the shelf’?
Response 1: Thank you for your comment. Accordingly, we have supplemented more explanations on the characteristics of NK cells as an “off-the-shelf” therapy product, especially about their safety. Please see the revised text from lines 55 to 61.
Point 2: Mention the benefits of the suicide gene- lane 286
Response 2: Thank you for your advice. We have added a description of the benefits of the suicide gene, such as controlling the number of infused NK cells to avoid toxic side effects. (lines 355 to 356).
Point 3: What does iCasp9 means in lane 308?
Response 3: Thank you for your advice. inducible caspase 9 (iCasp9) is a suicide gene, which can induce the rapid apoptosis of transduction cells. We have added relevant explanations in the revised text (lines 378 to 379).
Point 4: Mention what nanobodies are- lane312
Response 4: Thank you for your advice. Nanobodies are single immunoglobulin variable domains derived from camelids’ heavy-chain antibodies, which exhibit low immunogenicity, high specificity, and stability. We have added relevant explanations in the revised text (lines 382 to 384).
Point 5: What are the 4 different components? Lane 343
Response 5: Thank you for your comment. The 4 different components include DAP10, FcεRIγ, 2B4, and the ζ chain of the T-cell receptor. We have added relevant explanations in the revised text (lines 414 to 416).
Point 6: Explain what a master cell bank is- homo-haplotype?- lane 366
Response 6: Thank you for your comment. The main cell bank was established by selecting HLA homozygous iPSC clones. We have added relevant explanations in the revised text (lines 439 to 440).
Point 7: The figure shown is very comprehensive, in a way that may be too much information in one figure, making part B and C very confusing, as there is too much information that does not clearly matches the figure, for example, what is the triangle is meant to mean in 1B? decrease of something? 1B and 1 C, can be simplify in this figure (basic one), and made 2 more figures that show more information on B and C.
Response 7: Thank you for your advice. We have simplified the original Figure 1 according to your suggestion (Figure 2), and we have made Figure 3 and Figure 4 from the original contents of Part B and Part C. All patterns that appear are annotated.
Point 8: A figure of the general regulation of NKs activation, it will be very useful, and adding the modifications done on it.
Response 8: Thank you for your advice. We have made a new figure to overview the general regulation of NK cell activation (Figure 1). However, to avoid duplication, the relevant gene modifications are specifically described in Figure 3 and Figure 4, and we hope the correction will meet with approval.
Point 9: When speaking about different experimental work, the complexity of the models is great, so describing them is complex. The authors added almost in all cases how (methodology) the cells were generated (lentiviral, retoviral and so on). This information is not vital for the message, complicates the sentences and is already present in the Tables.
Response 9: Thank you for your advice. Accordingly, we have simplified or deleted the description of the methodology in the revised text as much as possible.
Point 10: Table 1, can be divided into solid tumors vs hematological ones
Response 10: Thank you for your advice. We have divided Table 1 into solid tumors and hematological tumors in the revised manuscript.
Point 11: Lane: 258 typically applied , will be best used
Response 11: Thank you for your advice. We have changed “applied” to “used” according to your suggestion (line 297).
Point 12: When mentioning a gene or gene product, please use scientific nomenclature CISH no CISH
Response 12: Thank you for pointing out this problem. We have carefully checked the manuscript and standardized all genes according to scientific nomenclature.
Point 13: Disolved ? lane 496
Response 13: Thank you for your advice. We have deleted this sentence: “This chimeric costimulatory converting receptor (CCCR) could bind PD-1 ligands to switch the negative signal to a positive one by activating NKG2D and 41BB signaling.” in the revised manuscript.
Point 14: Even lane 579 is not needed
Response 14: Thank you for your advice. We have deleted this sentence: “Moreover, modification of 4-1BB and CD3ζ further enhanced the anti-tumor activity of CD16-NK-92 and mediate stronger perforin and granzyme secretion.” in the revised manuscript.

Reviewer 4 Report
In this review, Xiao et al. present a comprehensive description of natural killer (NK) cells as relatively new tools for cancer immunotherapy. The paper is written in a clear manner and has a coherent structure. There are problems with English language, but for the rest, very few major issues or concerns have been identified.
MAJOR POINTS:
1) Lines 98-100: HLA-E molecules are also HLA class I molecules and should be labelled as such (even if non classical)
2) Lines 168 and 481-482: B7-H3 and HLA-G are not inhibitory receptors but inhibitory ligands (for NK cell inhibitory receptors)
3) 2B4 is NOT a NK cell-specific receptor
4) The reference list is presented twice
MINOR POINTS:
1) Lines 52-54: the affirmation "back into the patient" is only valid for autologous NK cells
2) Line 115: extracellular vesicles
3) Line 148: natural; there are other typos and mistakes in English language throughout the manuscript; therefore a careful editing is recommended
4) Line 367: please explain better the term "NK/ILC" cells in this context
5) Line 564: Bruce Walcheck's team...
6) In Tables 1 and 2, by far not all abbreviations used are explained but should be for a better understanding
Author Response
Dear Reviewer:
Thank you for your comments concerning our manuscript entitled “Natural Killer Cells: A Promising Kit in the Adoptive Cell Therapy Toolbox” (ID: cancers-1986173). Those comments are all valuable and very helpful for revising and improving our paper. We have studied the comments carefully and have made corrections which we hope meet with approval. We have provided a point-by-point response to your comments. Please see the attachment.
Response to Reviewer 4 Comments
Reviewer #4: In this review, Xiao et al. present a comprehensive description of natural killer (NK) cells as relatively new tools for cancer immunotherapy. The paper is written in a clear manner and has a coherent structure. There are problems with English language, but for the rest, very few major issues or concerns have been identified.
Reply: Thank you very much for reviewing our manuscript and giving your valuable comments.
MAJOR POINTS:
Point 1: Lines 98-100: HLA-E molecules are also HLA class I molecules and should be labelled as such (even if non classical)
Response 1: Thank you for pointing out this problem. We have changed the expression according to your suggestion. Please see the revised text from line 111.
Point 2: Lines 168 and 481-482: B7-H3 and HLA-G are not inhibitory receptors but inhibitory ligands (for NK cell inhibitory receptors)
Response 2: Thank you for pointing out this error. It's a very important piece of information. We confused the two concepts and have now corrected them. Please see the revised text from lines 325 to 326.
Point 3: 2B4 is NOT a NK cell-specific receptor
Response 3: Thank you for pointing out this error. We have changed the “NK cell-specific costimulatory receptor” into “NK-related costimulatory receptor” in line 400.
Point 4: The reference list is presented twice
Response 4: Thank you for pointing out this error. We are sorry for our carelessness. We have deleted the repeated reference list.
MINOR POINTS:
Point 1: Lines 52-54: the affirmation "back into the patient" is only valid for autologous NK cells
Response 1: Thank you for pointing out this problem. As an “off-the-shelf” therapy product, one of the most attractive advantages of NK cells is that patients can accept the infusion from allogeneic sources. To be better understood, we have changed "back into the patient" to "back into patients". (line 65)
Point 2: Line 115: extracellular vesicles
Response 2: Thank you for pointing out this error. We have changed “extracellular vesicle” into “extracellular vesicles”. (line 142)
Point 3: Line 148: natural; there are other typos and mistakes in English language throughout the manuscript; therefore a careful editing is recommended
Response 3: Thank you for pointing out this error. We have corrected this typo (line 211). And we acknowledge that there are grammar issues in the initially submitted manuscript after carefully reading it. We have corrected all these issues that we found in the resubmitted version.
Point 4: Line 367: please explain better the term "NK/ILC" cells in this context
Response 4: Thank you for your advice. Given that NK cells belong to the innate lymphoid cells (ILCs) family, we decide to delete ILC here to avoid ambiguity.
Point 5: Line 564: Bruce Walcheck's team...
Response 5: Thank you for pointing out this error. We apologize for our carelessness, and we have corrected this mistake (line 631).
Point 6: In Tables 1 and 2, by far not all abbreviations used are explained but should be for a better understanding
Response 6: Thank you for your advice. According to your suggestion, we have explained all abbreviations in Table 1 and 2.
